# Progress of Drug Nanocrystal Self-Stabilized Pickering Emulsions: Construction, Characteristics In Vitro, and Fate In Vivo

**DOI:** 10.3390/pharmaceutics16020293

**Published:** 2024-02-19

**Authors:** Jifen Zhang, Fangming Dong, Chuan Liu, Jinyu Nie, Shan Feng, Tao Yi

**Affiliations:** 1College of Pharmaceutical Sciences, Southwest University, Chongqing 400716, China; zhjf@swu.edu.cn (J.Z.); fengshan@swu.edu.cn (S.F.); 2Chengdu Institute of Food Inspection, Chengdu 611130, China; liuchuanai@hotmail.com; 3Faculty of Health Sciences and Sports, Macao Polytechnic University, Macau 999078, China

**Keywords:** Pickering emulsion, drug delivery, nanocrystals, characteristic, fate in vivo

## Abstract

A drug nanocrystal self-stabilized Pickering emulsion (DNSPE) is a novel Pickering emulsion with drug nanocrystals as the stabilizer. As a promising drug delivery system, DNSPEs have attracted increasing attention in recent years due to their high drug loading capacity and ability to reduce potential safety hazards posed by surfactants or specific solid particles. This paper comprehensively reviews the progress of research on DNSPEs, with an emphasis on the main factors influencing their construction, characteristics and measurement methods in vitro, and fate in vivo, and puts forward issues that need to be studied further. The review contributes to the advancement of DNSPE research and the promotion of their application in the field of drug delivery.

## 1. Introduction

The initial observation of solid particulates adsorbing at the interface of water and oil was reported by Ramsden in 1903 [1]. Pickering demonstrated in 1907 that ultrafine solid particles could be stably adsorbed at the water/oil interface, effectively preventing oil (or water) droplets in the dispersed phase from coalescence and aggregation [2]. Accordingly, this kind of emulsion, stabilized by the fine solid particles, was named a Pickering emulsion. Early Pickering emulsions were mostly stabilized by inorganic particles such as silica [3], calcite [4], calcium carbonate [5], iron oxide [6], hydroxyapatite [7], etc. However, the biocompatibility and biodegradability of inorganic particulates are poor, which raises safety concerns regarding their use in drug delivery systems. Later, researchers explored solid particles derived from food sources as stabilizers of Pickering emulsions [8,9]. A variety of macromolecule organic particles, such as protein [10,11], starch [12], cellulose [13], chitosan [14], etc., were successfully employed to stabilize Pickering emulsions, improving the safety of the emulsions. Many bioactive components, such as β-carotene [15,16], curcumin [17,18], rutin [19], canthaxanthine [20], bupivacaine [21], vitamin D3 [22], hesperidin [23], resveratrol [24], and myricetin [25], were delivered via Pickering emulsions stabilized by these macromolecule organic particles by dissolving drugs in the oil phase with the aim of improving oral absorption, controlling release rates, or enhancing the stability of drugs in the gastrointestinal tract [26,27].

Apart from the macromolecule organic particles of proteins, polysaccharides, fats, and cellulose, some small-molecule substances obtained from food sources, such as curcumin [28], glycyrrhizic acid [29], diosgenin [30], luteolin [31], dihydromyricetin [32], ursolic acid [33], and ginkgo biloba extracts [34], have also been used to stabilize Pickering emulsions. Most of these small-molecule compounds possess specific pharmacological activities. Numerous reports have documented the use of nanocrystals of bioactive constituents as stabilizers in Pickering emulsions, as shown in Table 1.

Recently there has been a growing interest in the utilization of poorly water-soluble bioactive particles as solid stabilizers for Pickering emulsions [35]. This new kind of Pickering emulsion can be called a drug nanocrystal self-stabilized Pickering emulsion (DNSPE). Compared with other emulsions, DNSPEs exhibit superior safety due to the absence of surfactants or other xenobiotic solid particles. In addition to dispersing in oil droplets, insoluble drugs can be adsorbed as nanocrystals on the surface of a DNSPE, resulting in a greater drug loading capacity. These advantages render DNSPEs promising drug delivery systems. They has been successfully used to deliver a diverse range of insoluble pharmaceuticals by oral and, more recently, topical and even injection administration.

As a novel Pickering emulsion, there remain many substantial issues about DNSPEs that require further investigation. This article provides a comprehensive overview of the current research on DNSPEs as drug delivery systems, analyzes in depth the primary factors influencing the construction of DNSPEs, the characterization methods of the structures and properties of DNSPEs in vitro, and the fate of DNSPEs in vivo. It further anticipates the most important future research directions and industrialization prospects of DNSPEs.

**Table 1 pharmaceutics-16-00293-t001:** Current reports of Pickering emulsions stabilized by small molecular bioactive particles.

Active Ingredient	Pharmaceutical Properties	Size and Potential of Nanocrystals	Oil	Preparation Method	Emulsion Type	Size and Potential of Emulsion Droplets	Drug Delivery Properties	Reference
Silybin	A component mainly used for the prevention and treatment of liver diseases with poor oral bioavailability due to its low water solubility.	232 ± 13 nm, −34.7 ± 0.3 mV	Glyceryl monocaprylate	Silybin nanosuspension was prepared first and then mixed with oil, followed by processing through a high-pressure homogenizer.	O/W	35.8 ± 4.2 μm ^a^, −31.4 ± 0.5 mV	The oral bioavailability of silybin was improved by DNSPE: its AUC increased by 1.6-fold and 4.0-fold compared with nanocrystals and coarse powder, respectively.	[36]
Curcumin	A natural polyphenolic bioactive compound with many various health benefits; low oral bioavailability due to its low water solubility and permeability.	220 nm	High-oleic sunflower oil	Curcumin nanosuspension was prepared first and then added to the oil phase. The mixture was emulsified using ultra-probe sonication.	O/W	1.2 μm ^b^	Nano-sized amorphous curcumin particles fabricated using nanonization technology could be used to stabilize the Pickering emulsion.	[28]
Curcumin		126.20 ± 1.16 nm, −29.50 ± 0.67 mV	Labrafil M 1944 CS:curcuma aromatic oil = 1:1	Curcumin nanosuspension and oil phase were mixed and cut at 22,000 rpm for 10 min, followed by homogenization.	O/W	163.66 ± 6.78 nm ^g^ −36.41 ± 0.56 mV	DNSPE improved the oral bioavailability of curcumin by improving drug release and promoting transport across Caco-2 cells, and enhanced its therapeutic effect in airway inflammation.	[37]
Curcumin		Dispersed in oil: 0.2 μm, −47.6 ± 2.4 mV	Soybean oil of 95% wt %	Polyphenol was dispersed into soybean oil first. Aqueous phase was mixed with this oil phase, followed by passing through a high-pressure Leeds jet homogenizer.	W/O	9−10 μm ^d^	W/O emulsions could be stabilized by crystals from naturally occurring polyphenols, which may lead to various soft matter applications.	[38]
Quercetin	A plant flavonoid with a variety of effects such as anti-inflammatory, antioxidant, anti-hypertensive. But it has low solubility, chemical instability, and low bioavailability.	354.88 ± 17.35 nm	Labrafac Lipophile WL 1349	Quercetin nanosuspension was prepared first and then oil was added and mixed by an ultrasonic cell grinder.	O/W	10–11 μm ^a^, −35 mV	NSSPE is a promising oral delivery system for improving the oral bioavailability of quercetin: its AUC_0–t_ increased by 2.76 times and 1.38 times compared with coarse powder and nanocrystals.	[39]
Quercetin		130 nm	Soy oil	The quercetin dispersions were mixed with soy oil at different ratios. The mixture was by homogenized at 10,000 rpm for 2 min.	O/W	95 μm ^g^	Natural quercetin treated with the antisolvent method had a good ability to stabilize a Pickering emulsion, and this emulsion may have good prospective application potential for the development of novel and functional emulsion foods.	[40]
Ursolic acid	A typical pentacyclic triterpene carboxylic acid whose potential application as a therapeutic agent was hindered by its poor water solubility and permeability.	2–20 µm	Rapeseed oil	Ursolic acid was dispersed in oil first and put at 80 °C for 20 min. Water was then added and mixed using a high-speed homogenizer.	W/O	6–7 μm ^c^	Provided a new strategy for the construction of a carrier-free delivery system for pentacyclic triterpenes.	[33]
Diosgenin	A natural steroidal sapogenin with favorable biological activity, such as the prevention and treatment of cancer, cardiovascular diseases, and diabetes.		Canola oil	Diosgenin was dispersed in oil at 120 °C. Then, 80 °C preheated water was added and mixed. The mixture was homogenized.	Gel-like W/O	34.67 ± 0.58 μm ^c^	Diosgenin was a novel Pickering emulsifier to stabilize W/O emulsions; it had great potential for application in the food, cosmetic, and pharmaceutical fields.	[30]
Dihydromyricetin	A bioactive component of Ampelopsis grossedentataleaves.	Up to 8 μm in length	Medium-chain triglyceride	Oil and DMY aqueous dispersions were mixed and then homogenized using an IKA Ultra-Turrax T18 disperser.	O/W	20–30 μm ^a^	Dihydromyricetin could spontaneously and quickly transfer to the oil–water interface, reduce the interfacial tension, and enhance the interface thickness to stabilize Pickering emulsion gels.	[32]
Puerarin	A potential therapeutic agent for cerebrovascular diseases with low solubility and low permeability.	390.9 ± 78.5 nm, −44.7 ± 3.0 mV	Ligusticum chuanxiong oil and Labrafil M 1944 CS (9:1, *v*/*v*)	Puerarin nanosuspension was prepared first and mixed with oil, then processed through a high-pressure homogenizer.	O/W	12.9 ± 2.9 μm ^f^, −44.7 ± 2.2 mV	The oral bioavailability of puerarin was improved: the relative bioavailability of NSSPE compared to coarse powder, nanocrystals, and surfactant emulsion was 262.43%, 155.92%, and 223.65%, respectively.	[41]
Puerarin, ferulic acid, salvianolic acid B, and tanshinone II A	Main active components of Tongmai prescription. All had poor oral biavailability due to their low water solubility or peameability.	0.39 ± 0.01 μm −59.70 ± 0.50 mV	Ligusticum chuanxiong essential oil and Labrafil M 1944 CS (9:1, *v*/*v*)	Drug nanosuspension was prepared first and then oil was added and mixed, followed by processing through a high-pressure homogenizer.	O/W	1.43 ± 0.09 μm ^d^, −54.17 ± 0.85 mV	DNSPE was a promising oral drug delivery system for complicated traditional Chinese medicine, promoting the absorption of various components in NSSPEs.	[42]
Sinomenine	Sinomenine has analgesic, anti-inflammatory, and immunomodulatory effects with many shortcomings, such as low oral bioavailability, short biological half-life, severe adverse reactions.	121.49 ± 18.26 nm	Medium-chain triglyceride	Sinomenine nanosuspension was prepared first and then oil was added and emulsified.	O/W	1086.70 ± 66.30 nm ^d^, −25.4 ± 0.3 mV	Intra-articular injection of sinomenine DNSPE had a better therapeutic effect on rheumatoid arthritis in rats and reduced the toxic and side effects of sinomenine than sinomenine suspension.	[43]
Luteolin	A typical polyphenolic substance derived from natural plants with good antioxidant activity.	379.3 ± 46.14 nm −19.36 ± 2.12 mV	Pine nut oil	The oil was poured into the luteolin micro-nanosuspension and subjected to high-speed homogenization.	O/W	125.6 ± 24.7 nm ^g^, −26.21 ± 3.63 mV	The dense protective layer that luteolin micro-nano particles formed stabilized the Pickering emulsion. The particles also improved the oxidation stability of pine nut oil.	[31]
*Quercus suber* bark (QSB) from *Quercus suber* L.	A natural lightweight material with characteristics such as being impermeable to liquids, good biodegradablility, antioxidant, anti-inflammatory, radical scavenger, and antimicrobial properties.	47.4 ± 0.1 μm	Caprylic/capric acid triglyceride	QSB particles were dispersed in oil first. The oil and aqueous phases were then mixed.	O/W	87.3 *±* 1.0 μm ^e^	The use of QSB particles as a multifunctional solid ingredient contributed to achieving a stable, effective, and innovative Pickering emulsion with a meaningful synergistic protection against oxidative stress.	[44]
Tiliroside	A natural flavonoid which is relatively insoluble in water with log_10_ P of 2.71.	Close to 500 nm −28 mV	n-tetradecane	Flavonoid dispersions were prepared and then mixed with oil, followed by passing through a high-pressure-jet homogenizer.	O/W	About 100 μm ^d^	The ability of tiliroside to act as a Pickering emulsifier is sensitively dependent on pH, as a result of changes in the solubility as a function of pH.	[45]
Rutin hydrate	Two flavonoids which are edible, commercially attractive, and linked to particular health benefits.	In oil: 10.52 ± 0.57 μm In water: 0.18 ± 0.01 μm	Sunflower oil	Particles were sonicated in the continuous phase and heated to 45–50 °C. Then, a dispersed phase was added to them and emulsified.	O/W and W/O	O/W: about 25 μm ^d^ W/O: about 12 μm ^d^	Provided a series of clean label food-grade emulsions in the food industry. The main particle prerequisites for successful Pickering stabilization were: particle size (200 nm–1 μm), an affinity for the emulsion continuous phase, and a sufficient particle charge.	[46]
Naringin		In oil: 18.64 ± 0.82 μm In water: 6.41 ± 0.41 μm				O/W: about 50 μm ^d^ W/O: about 55 μm ^d^		
Ginkgo biloba extracts (GBEs)	GBE has strong free radical scavenging activity, and antioxidant and antiatherosclerosis capacities due to the synergistic effects of flavonoids and triterpene lactones. However, the poor liposolubility and water solubility of flavonoids resulted in low oral bioavailability.		Soybean oil	FA or GBE powder was dispersed in water using an ultrasonic processor and then mixed with oil. The mixture was prehomogenized using a high-speed homogenizer and then passed through a microfluidizer.	O/W	0.51 μm ^d^ 0.92 μm ^d^	Potential applications of FA and GBE particles in the design of food emulsions aiming at controlling the lipid digestion rate and at increasing the bioaccessibility of foods containing flavonoids.	[34]
Flavonoid glycosides fraction (FA)					O/W	0.51 μm ^d^ 0.92 μm ^d^		
Ginkgolide B (GB)	A natural diterpene with notable pharmacological attributes, including neuroprotective, antioxidant, and anti-inflammatory effects. The low solubility of GB in gastric acid and poor stability in intestinal fluid significantly impede its oral absorption.	85.4 ± 1.4 nm, −11.8 ± 0.6 mV	Ligusticum chuanxiong essential oil	GB nanocrystals were prepared by a downsized wet bead grinding technique first. Then, oil was added to the dispersion and it was ground again at 1000 rpm for 4 h.	O/W	629.5 ± 19.3 nm, −22.7 ± 0.9 mV	GB-DNSPNE, with its intact nanoparticle slow release and absorption, was more effective in enhancing the oral bioavailability and anti-ischemic stroke efficacy of GB compared to the rapid release and absorption of GB-NCs.	[47]

Note: ^a^ mean size calculated based on optical images; ^b^ *d*_4,3_ measured by Malvern Mastersizer; ^c^ *d*_3,2_ calculated based on optical images; ^d^ *d*_3,2_ measured by a laser particle analyzer; ^e^ *d*_0.5_ measured by Malvern Mastersizer; ^f^ *d*_4,3_ calculated based on optical images; ^g^ mean particle size measured by Zeta-PALS analyzer.

## 2. Materials and Methods

Studies reported from 2011 to 2023 were searched on the databases ‘Web of Science (http://www.webofknowledge.com)’ accessed on 1 November 2023, ‘PubMed (http://pubmed.cn/)’ accessed on 1 September 2023, and ‘CNKI (https://cnki.net/)’ accessed on 30 June 2023 using ‘Pickering emulsions’ as a topic. A subset of significant reviews and articles pertaining to Pickering emulsions stabilized by small molecular bioactive particles were chosen from a vast collection of articles. This review was written based on an analysis of these selected literatures.

## 3. Current Application of DNSPEs in Drug Delivery

### 3.1. Oral Administration of DNSPEs

The first drug delivery system based on a DNSPE was reported by Yi; it was used to improve the oral bioavailability of silybin [36]. Silybin is used to treat acute and chronic hepatitis, cirrhosis, and toxic liver injury in clinics due to its beneficial effects in protecting normal liver cells and helping damaged liver cells to recover. However, the oral bioavailability of silybin is low owing to its poor water solubility and membrane permeability. A Pickering emulsion stabilized by silybin nanocrystals (SN-DNSPE) was prepared by mixing silybin nanocrystals of about 300 nm with Capmul C8 by high-pressure homogenization. The resultant emulsion droplet size was 27.3 μm. SN-DNSPE was much more stable than the nanocrystal suspension of silybin. It remained stable at room temperature for more than 40 days, whereas the nanocrystal suspension began to layer after only 4 days. The AUC and the C_max_ of silybin in the DNSPE were 4.0 and 1.6 times and 3.6 and 2.5 times greater than those of the crude material suspension and nanocrystal suspension, respectively, following gavage administration to rats, indicating that the DNSPE significantly improved the oral absorption of silybin and the improvement was even better than that of the nanocrystal suspension.

Up to now, DNSPEs have been employed to deliver many other poorly water-soluble drugs orally, such as curcumin, quercetin, ginkgolide B, puerarin, and even traditional Chinese medicinal compounds. Curcumin is an effective anti-inflammatory agent extracted from turmeric’s rhizome. However, its clinical applications have been limited by its extremely low oral bioavailability. A curcumin nanocrystal self-stabilized Pickering emulsion (Cur-DNSPE) with a droplet size of 163.66 ± 6.78 nm was prepared [37]. The cumulative release of curcumin in vitro was enhanced, and the transport of curcumin across Caco-2 cells was promoted, which led to enhanced oral bioavailability and anti-inflammatory effects in rats. The expression of inflammatory factors NO, IL-6, TNF-a, MDA, IgE, and ICAM-1 was inhibited and the expression of IL-10 and SOD was improved significantly.

Quercetin exhibits a variety of pharmacological activities, including anti-inflammatory, antioxidant, anti-hypertensive, and neuroprotective. However, its solubility in water is only 0.01 mg·mL^−1^, restricting its oral absorption. Wang et al. [39] developed a DNSPE stabilized by quercetin nanocrystals with a drug loading of 5.35 mg·mL^−1^ using Labrafac Lipophile WL 1349 as the oil phase. The drug release and oral absorption of the DNSPE significantly improved compared with crude material and nanocrystal suspension of quercetin. After 24 h, the cumulative drug release rate of the DNSPE in phosphate buffer (pH = 7.4, containing 1% sodium dodecyl sulfate) was 68.88%, surpassing the rates of 20.15% for the crude material and 50.71% for the nanocrystals. The AUC_0–t_ of the DNSPE after gavage in rats was 2.76 and 1.38 times greater than those of the crude material and nanocrystals, respectively, and the C_max_ was 2.4 and 1.4 times greater.

Compared with traditional emulsions stabilized with surfactants, drugs in DNSPEs are not only dissolved in the oil phase but also adsorbed at the oil–water interface of the droplets. Differences in the structure lead to different drug loading and dissolution kinetics. It seems that the improvement in drug loading and dissolution might be related to (1) some crystals dissolving in the oil phase of the emulsions, (2) increasing the surface area of the drug crystals through homogenization, and (3) enhanced exposure of particles due to interlocking at the interface that affects their kinetics of dissolution. The clear mechanism remains to be further explored.

The above DNSPEs mostly employed medium-chain fatty acid glycerides or polyethylene glycol glycerol oleate as the oil phase. Given the abundance of insoluble and volatile pharmacodynamic components in many traditional Chinese medicine (TCM) formulations, the idea of DNSPEs utilizing mixed nanocrystals of different active compounds from TCM formulas as solid particles and the volatile oil as an oil phase was proposed. The initial successful application of the idea was for puerarin. Puerarin has favorable pharmacological effects, such as the inhibition of atherosclerosis, improvement of microcirculation, anti-myocardial ischemia, and anti-arrhythmia, etc. However, it belongs to class IV according to the BCS classification system, and its oral bioavailability in rats is limited. Chuanxiong Rhizoma has the effects of dilating blood vessels, improving microcirculation, and relieving pain, and is frequently used in the treatment of many cardiovascular and cerebrovascular diseases. Volatile oil is one important active component of Chuanxiong Rhizoma. With Ligusticum chuanxiong oil and Labrafil M 1944 C (9:1, *v*/*v*) as an oil phase, a DNSPE stabilized by puerarin nanocrystals was developed [41]. A study on its oral bioavailability in rats showed that the DNSPE significantly enhanced the oral absorption of puerarin. The AUC was increased by 1.6, 0.56, and 1.24-fold, respectively, compared with the crude material, nanocrystals, and emulsions stabilized by Tween 80.

Ginkgolide B, a naturally occurring platelet-activating factor antagonist, is widely used for the treatment of cardiovascular and cerebrovascular ailments. Its limited oral bioavailability, however, impedes its practical application. A ginkgolide B nanocrystal self-stabilized Pickering emulsion (GB-DNSPE) was prepared using a miniaturized wet bead milling method with Ligusticum chuanxiong oil as the oil phase [47]. In contrast to Ginkgolide B nanocrystals, GB-DNSPE demonstrated superior efficacy in enhancing the oral bioavailability of GB due to the slow release and absorption of the intact nanoparticle. The relative oral bioavailability of GB-DNSPE was approximately 5.96 times and 1.63 times greater than crude GB powder and GB nanocrystals, respectively. Hence, the anti-ischemic stroke efficacy of GB was significantly improved by the DNSPE.

DNSPEs have even been used for TMC formulas containing multiple medicinal ingredients. Tongmai is a traditional TCM remedy with effects of promoting blood circulation and removing blood stasis. It is comprised of Puerariae lobatae radix, Chuanxiong rhizoma, and Salvia miltiorrhiza. Puerarin, ferulic acid, salicylic acid B, tanshinone IIA, and Ligusticum chuanxiong oil are the primary pharmacodynamic components. A DNSPE with the Tongmai formula was prepared using a nanocrystal mixture comprising puerarin, ferulic acid, salicylic acid B, and tanshinone IIA as an aqueous phase, and a mixture of Ligusticum chuanxiong oil and Labrafil M 1944 C (9:1, *v*/*v*) as an oil phase [42]. The DNSPE exhibited superior physical stability in comparison to nanocrystal suspensions due to the strong adsorption film formed by the adsorption of 15.40% of the puerarin, 15.39% of the ferulic acid, 10.97% of the salvianolic acid B, and 31.51% of the tanshinone IIA onto the surface of the Ligusticum chuanxiong oil droplets. The results of the Caco-2 cell monolayer transport experiment indicated that the apparent permeability coefficient (P_app_) from the apical side to the basal side (AP → BL) increased 1.21-fold and 0.93-fold for puerarin and salicylic acid B, respectively, when compared with the unprocessed crude material dispersion. Crude tanshinone had a negligible capacity to traverse the Caco-2 cell monolayer, whereas its Papp_AP → BL_ in the DNSPE increased to 5.78 × 10^−6^ cm·s^−1^. Puerarin, ferulic acid, salicylic acid B, tanshinone IIA, and Ligusticum chuanxiong oil were all classified as both ‘excipients’ and ‘drugs’ in the DNSPE, potentially exhibiting synergistic effects. This kind of DNSPE is particularly promising for oral preparations of TCM formulas containing complex active compounds.

### 3.2. Injectable Drug Delivery of DNSPEs

Up to now, there have only been a few studies on the injection administration of Pickering emulsions. These injectable Pickering emulsions have been primarily investigated for subcutaneous delivery of vaccines and immune adjuvants [48,49,50] utilizing PLGA-based nanoparticles, such as Chinese yam polysaccharide PLGA nanoparticles and Lentinan PLGA nanoparticles as solid particle stabilizers. An intra-articular DNSPE was reported recently. Sinomenine has anti-inflammatory, analgesic, and immunomodulatory effects, and is used for the treatment of rheumatoid arthritis by oral tablet or intramuscular injection clinically. The current preparations have many defects, including low oral bioavailability, short biological half-life, severe gastrointestinal adverse reactions, serious damage to liver and kidney, and cardiotoxicity. In order to over these shortcomings, a sinomenine DNSPE injection was prepared using sinomenine nanocrystals as a stabilizer and medium-chain fatty acid triglycerides as an oil phase [43]. In contrast to the intragastric or intra-articular administration of the sinomenine suspension, intra-articular injection of the DNSPE could effectively reduce joint swelling and the inflammation index, improve synovial tissue lesions, and inhibit joint inflammation. The DNSPE also showed a superior therapeutic effect compared with the sinomenine suspension, because it could form a drug reservoir after injection into the joint cavity, thereby releasing the drug slowly and stably.

### 3.3. Topical Drug Delivery of DNSPEs

Catarina et al. [44] developed a Pickering emulsion with natural green *Quercus suber* bark (QSB) particles as the stabilizer. The QSB solid particles were dispersed first in caprylic/capric triglycerides and then mixed with pure water using a high-speed shearing machine to produce an emulsion with an approximate particle size of 90 μm. Devoid of surfactants or additional solid particles, the emulsion comprised solely the oil phase, pure water, and QSB solid particles; thus, it was classified as a Pickering emulsion. QSB is a lightweight material obtained from the outer bark of *Quercus suber* L., consisting of suberin, lignin, cellulose, extractives, a small amount of fatty acids, terpenes, long-chain aliphatic compounds, and saccharides. This Pickering emulsion stabilized by QSB particles was a shear-thinning fluid, and was non-irritating to skin, which made it especially suitable for topical administration. QSB possesses specific antimicrobial, antioxidant, and anti-aging activities, which provided the Pickering emulsion with certain pharmacological effects, including protection against oxidative damage in the HaCaT cell line and inhibition of human neutrophil elastase.

## 4. Main Factors Influencing the Construction of DNSPEs

According to current research, the wettability, particle size, charge and concentration of drug nanocrystals, preparation methods, and volume ratio of the oil phase to water phase are the primary determinants influencing the preparation and stability of DNSPEs.

### 4.1. Wettability of Nanocrystals

The wettability of the solid particles plays a crucial role in determining the type and stability of formed Pickering emulsion [51,52]. In order to quantify the wettability of the particles, the three-phase contact angle (*θ*) among the solid particles, oil, and aqueous phase is typically utilized. Generally, particles with *θ* < 90° are highly hydrophilic and mainly wetted by the aqueous phase, forming oil-in-water (O/W) emulsions. Conversely, particles with *θ* > 90° are highly hydrophobic and mainly wetted by the oil phase, forming water-in-oil (W/O) emulsions (see Figure 1). According to Kaptay [53], in the case of O/W emulsions stabilized by a single layer of particles, the contact angle must be 15° < *θ* < 90°. Similarly, for W/O emulsions, the contact angle must be 90°< *θ* < 165°. For emulsions stabilized by a double layer of particles, O/W emulsions remain stable for contact angle values of 15° < *θ* < 129.3°, whereas W/O emulsions remain stable for contact angle values of 50.7° < *θ* < 165°.

Usually, Pickering emulsions have better stability against coalescence than surfactant-stabilized emulsions due to the relatively rigid adsorption of solid particles at the interface of the emulsion droplets [54]. The separation energy required for desorption of perfectly spherical solid particles from the monolayer adsorption film at the oil/water interface is expressed by Equation (1).
Δ*E* = π*r*^2^*γ*_ow_(1 − |cos *θ*_ow_|)^2^(1)
where Δ*E* represents the desorption energy, *r* represents the radius of the spherical solid particles, *γ* represents the interfacial tension of water and oil, and *θ*_ow_ represents the three-phase contact angle of the solid particles.

In general, within the *θ* range of 30°–150°, the desorption energy far exceeds the thermal energy of Brownian motion. Consequently, the solid particles will be firmly adsorbed at the oil–water interface, resulting in the formation of a stable Pickering emulsion. The closer *θ* is to 90°, the higher the desorption energy is and the more stable the Pickering emulsion is [55]. The hypothetical equation holds for perfectly spherical particles. Non-spherical particles of the same volume and material tend to have a larger specific surface area and will have a higher desorption energy [38].

A DNSPE is a Pickering emulsion in essence, so the aforementioned rule might remain valid. The *θ* of puerarin in Ligusticum chuanxiong oil and Labrafil M 1944CS was 82.14° and 14.80°, respectively. Hence, the DNSPE of puerarin prepared with Ligusticum chuanxiong oil as an oil phase exhibited a good stability, whereas no stable emulsion could be obtained with Labrafil M 1944CS as an oil phase [56]. The *θ* of dihydromyricetin at the medium-chain triglyceride–water interface was 118.8°, so an O/W Pickering emulsion was formed [32]. The *θ* of ursolic acid at the canola oil–water interface was 156.4°, so it was more likely to form a stable W/O emulsion [33].

### 4.2. Solid Particle Size

The solid particle size is another important factor affecting the formation and stability of Pickering emulsions. Generally speaking, the specific surface area of solid particles increases with decreasing particle size, which is advantageous for enhancing their adsorption at the oil–water interface. With decreasing solid particle size, the film density at the interface of oil and water increases, which further strengthens the rigid barrier effect and promotes the stability of Pickering emulsions. It was reported that the size of solid particles should be at least one order of magnitude smaller than that of the emulsion droplets to form a stable Pickering emulsion [57]. At present, Pickering emulsions are predominantly stabilized by macromolecule organic particles of protein, cellulose, and starch, and so on. The size of the solid particles is mostly hundreds of nanometers, and the size of the emulsion droplets is generally from a few micrometers to tens of micrometers [58,59].

For DNSPEs, a similar relationship was observed between the solid particle size and the emulsion droplet size. For example, the average sizes of DNSPE droplets for silybin [36], puerarin [41], sinomenine [43], and ginkgolide B [47] were 27.3 μm, 12.4 μm, 1.58 μm, and 629.5 nm, respectively, and the average sizes of the corresponding drug particles were 339.5 nm, 390.9 nm, 121.49 nm, and 85.4 nm, respectively. The droplet size of the DNSPE was approximately 8–80 times the size of the corresponding drug particles. Overall, the nano-sized or submicron-sized particles of poorly soluble drugs made them easier to adsorb at the oil–water interface, thereby stabilizing the DNSPE. Drug nanocrystals are usually produced by dispersing the drug’s crude materials in the continuous phase, followed by high-pressure homogenization or probe ultrasonication. The particle size can be regulated by adjusting the homogenization pressure or ultrasonic power. Additionally, the pH of aqueous phase may also affect the particle size of poorly soluble drugs containing certain acidic or basic groups. Wang et al. [60] used puerarin and ferulic acid suspensions with different aqueous pH to prepare DNSPEs by high-pressure homogenization. When the pH of the aqueous phase was 5, the particle size of the suspensions reached 4 to 6 μm, preventing the formation of a stable emulsion. Conversely, when the pH of the aqueous phase exceeded 9, the particle sizes of puerarin and ferulic acid decreased to approximately 300 nm and 500 nm, respectively, producing stable DNSPEs. The absence of a statistically significant variation in the three-phase contact angles across various aqueous-phase pH values suggested that the micrometer size of the drug particles played a crucial role in the inability to form a DNSPE.

However, the relationship between the size of a DNSPE droplet and particle size might be more intricate. A curcumin DNSPE of 163.66 nm was prepared with curcumin submicron crystals of 126.20 nm [37]. Wang et al. [31] even obtained a stable DNSPE with an average droplet size of 125.6 nm by homogenizing luteolin submicron crystals of 379.3 nm and pine kernel oil under high pressure. In this instance, why the emulsion droplet size was smaller than the nanocrystal size deserves extensive investigation. One plausible hypothesis posits that the high-pressure homogenization process reduced the particle size of the luteolin nanocrystals, which might have led to the nanocrystals adsorbing onto the oil droplet surface at an even finer particle size. 

### 4.3. Charge of Nanocrystals

The droplet size and stability of DNSPEs are additionally affected by the ζ-potential of the nanocrystals. On the one hand, the adsorption of drug nanocrystals onto the surface of emulsion droplets should render the emulsion droplets an equivalent charge to that of the nanocrystals. An increase in the charge of the nanocrystals within a certain range leads to a corresponding increase in the charge of the emulsion droplets and a stronger electrostatic repulsion between the droplets, which is expected to prevent the droplets coalescing and improve the stability of the emulsion. On the other hand, the increase of charges exhibited by the nanocrystals might result in a correspondingly increased intercrystal repulsion, potentially impeding the adsorption of the nanocrystals at the interface of the emulsion droplets. Experimental verification is still necessary to validate this inference. Wang et al. [43] prepared a sinomenine DNSPE by introducing the oil phase into sinomenine nanosuspensions of varying pH values. At a pH of 4, the ζ-potential of the nanocrystals was −20.87 mV and the emulsion droplet size was 33.47 μm. As the pH value increased to 9, the ζ-potential of the nanocrystals gradually decreased to −32.73 mV and the emulsion droplet size gradually decreased to 17.24 μm due to the increase in charge repulsion between emulsion droplets.

The charge of nanocrystals may also influence their particle sizes. Luo et al. [45] prepared DNSPEs utilizing the nanocrystals of tiliroside, a flavonoid compound, as the stabilizer. As the aqueous-phase pH rose from 2 to 8, the charge of the nanocrystals decreased from 1.8 to −28 mV, which in turn affected the size of the nanocrystals. The reduction in the ζ-potential resulted in an initial marginal increase in the size of the nanocrystals, followed by a substantial decrease. The nanocrystal particle size peaked at over 1 μm at a ζ-potential of −3 mV, and then gradually decreased to 500 nm or less as the ζ-potential continued to decrease. Due to the dual effects of charge and particle size of nanocrystals, the droplet size of tiliroside DNSPE decreased from 150 μm to 30 μm when the aqueous pH rose from 2 to 8. Overall, there are few systematic studies on the effect of nanocrystal charge, and further exploration is required.

### 4.4. Nanocrystal Concentration

The formation and stabilization of DNSPEs depend mainly on the adsorption of nanocrystals on the surface of emulsion droplets. Consequently, the concentration of nanocrystals is another critical factor that influences not only the formation of a DNSPE but also the size of the emulsion droplets. An insufficient concentration of nanocrystals may lead to inadequate coverage of the emulsion droplet surface, resulting in a weak barrier effect around droplets and poor stability of the DNSPE. Yi et al. [36] demonstrated that when the quantity of oil was held constant, silybin nanocrystals of 100 mg or 200 mg could only cover part of the surface of the oil droplets, leading to a large emulsion droplet size and poor emulsion stability. When silybin nanocrystals reached 300 mg, the oil droplets were covered entirely by the nanocrystals, leading to a reduction in the emulsion droplet size and an enhancement in the stability of the DNSPE. The droplet size of a quercetin DNSPE also decreased as the nanocrystal concentration increased [40].

It should be noted that the maximum interfacial adsorption amount of nanocrystals is restricted by the finite total surface area of the emulsion droplets [61]. An overabundance of interfacial adsorption may result in the precipitation or formation of a three-dimensional network gel structure of nanocrystals. For example, the stability of the sinomenine DNSPE was improved with an increase in the nanocrystal concentration within the range of 1–5 mg/mL. However, once the nanocrystal concentration exceeded 20 mg/mL, the interfacial adsorption became saturated and the excess nanocrystals precipitated from the DNSPE [43]. Given that the adsorption of nanocrystals onto the surface of emulsion droplets is influenced by several variables, including wettability, nanocrystal particle size, charge, and total surface area of the emulsion droplets, the optimal nanocrystal concentration differs across various DNSPEs. For example, the recommended nanocrystal concentrations for DNSPEs of puerarin, quercetin, and curcumin were 1–5 mg/mL [41], 4–6 mg/mL [39], and 3–4 mg/mL [28], respectively.

### 4.5. Preparation Method

Generally, DNSPEs are prepared by mixing drugs, water, and oil, and followed by high-pressure homogenization or probe ultrasonication. According to a number of studies, the initial dispersion phase of drug solid particles has a substantial impact on the formulation of DNSPEs. Silybin DNSPEs could be obtained successfully via high-pressure homogenization as long as silybin was first dispersed in water before being mixed with the oil phase of Capmul^®^ MCM C8, regardless of whether silybin was in the form of nanocrystals or coarse particles of the raw material. In contrast, when silybin was initially dispersed in Capmul^®^ MCM C8 and then mixed with water for high-pressure homogenization, it aggregated into irregular aggregates in oil, and could not adsorb onto the surface of oil droplets to form a DNSPE [62]. Conversely, an array of outcomes was noted in the case of puerarin DNSPEs. If puerarin was dispersed in the oil phase ultrasonically first and subsequently mixed with water by high-pressure homogenization, a stable DNSPE could be obtained [63]. One potential explanation for this variation might be attributed to the wettability of the drugs. Silybin exhibits excellent solubility in the oil phase but is virtually insoluble in water due to its high hydrophobicity. In contrast, puerarin is hydrophilic and slightly soluble in water, while it is virtually insoluble in oil. Consequently, it is likely that drug particles ought to be initially dispersed in a phase characterized by low solubility. Further validation is needed for this speculation.

Furthermore, the majority of O/W emulsions were prepared by initially dispersing solid particles in the aqueous phase, whereas W/O emulsions were prepared by initially dispersing solid particles in the oil phase. Additionally, the preparation temperature was also different for O/W and W/O emulsions. To obtain a homogeneous oil phase, the O/W DNSPEs frequently necessitated heating the oil phase containing solid particles, whereas O/W DNSPEs were predominantly prepared at room temperature. For instance, when diosgenin [30] and ursolic acid [33] were used to stabilize W/O DNSPEs, the drug particles were initially dispersed in rapeseed oil heated at 120 °C for 30 min, or dispersed in canola oil heated at 80 °C for 20 min, respectively. Following this, the aqueous phase was added and mixed by a high-shear mixer.

A specific quantity of mechanical energy is necessary for the formation of stable emulsions, and the emulsification technique may also have an impact on the production and stability of DNSPEs. Duffus et al. [46] compared the effects of high-shear and high-pressure homogenization on O/W DNSPEs stabilized by two flavonoids, rutin hydrate (which is strongly hydrophilic) and naringin (which is hydrophobic). Naringin DNSPEs generated via high-shear mixing at 10,000 rpm for 2 min had a droplet size of approximately 60 μm, which increased to 90 μm after a storage period of 14 days. Additional high-pressure homogenization after high-shearing reduced the droplet size to approximately 0.2 μm, and the droplet size remained virtually unchanged after 14 days of storage. It was suggested that high-pressure homogenization reduced the emulsion droplet size and improved the stability of the DNSPE, which was consistent with the findings of the majority of studies [36,39,41]. Unexpectedly, the droplet size of the rutin DNSPE prepared by high shear was about 10 μm, and there was no substantial alteration in droplet size within 14 days. A further high-pressure homogenization actually increased the emulsion droplet size to about 50 μm and decreased the stability of the DNSPE [46]. A potential rationale for this anomaly could be that during the subsequent homogenization at high pressure, the already formed emulsion droplets broke up into smaller ones, leading to a rapid increase in the total surface area of the droplets. However, there was not enough adsorption of rutin particles, which ultimately resulted in the rapid aggregation of small emulsion droplets and the development of the larger ones [46]. Therefore, considering the various aforementioned factors affecting their construction, including the wettability, particle size, and charge and concentration of drug nanocrystals, it is imperative to choose a suitable homogenization method for the formulation of a particular DNSPE.

## 5. In Vitro Characteristics of DNSPEs

### 5.1. Contact Angle

The contact angle is determined using the sessile drop method, which is described in a number of publications [64]. The common procedure is that a drop of water or oil is added to the drug surface, images are captured following stabilization, and the contact angle is then computed using software. The operations of contact angle measurement and results reported in the literature are listed in Table 2.

It should be noted that there are still some minor yet significant differences in the precise operational particulars. In some studies, the tablet of the drug was exposed to air [30], whereas in other studies the tablet was submerged in the oil and allowed to equilibrate for a specified duration [32]. The contact angle in the former was actually the three-phase contact angle of drug–air–water, while in the latter it corresponded to drug–oil–water. Not only are the contact angles measured by these two methods different, they may even show different hydrophilic and hydrophobic properties. For example, the contact angle of ursolic acid was 100.3° in air–water, but 156.4° in rapeseed oil–water [33]. Ferulic acid exhibited hydrophilicity in air–water with a contact angle of 69.5°, and lipophilicity in Ligusticum chuanxiong oil–water with a contact angle of 145° [56]. In current studies, either crude materials of drugs [46] or lyophilized powder of nanocrystal suspensions [28] were pressed into tablets. The dissimilarities in characteristics between crude powder and lyophilized powder of nanocrystals may result in disparate contact angles. For example, the contact angle of native quercetin in air–water (120.65°) was significantly greater than that of quercetin nanocrystals. Specifically, the contact angle of nanocrystals ranging in size from 130 to 600 nm was reduced from 84.85° to 65.15° [40]. Considering that freeze drying may alter the surface properties of nanocrystals, some researchers recently produced a drug film about 100 μm thick by casting a nanosuspension directly into a Petri dish and drying it at room temperature. This film was utilized to quantify the contact angle of egg yolk low-density lipoprotein particles [65], chitin nanocrystals particles [66], and chitosan casein phosphopeptide nanocomplex particles [67].

The contact angle measurement results will be marginally affected by the measurement approach employed. Which method is the more rational or more consistent with the above Equation (1) to explain the relationship between the wettability and the stability of DNSPEs? This is still unknown. Up to now, there are no systematic studies on this issue. It is difficult to come to a conclusion by analyzing the reported studies due to variations in the oil phase, nanocrystal particle size, preparation method, and oil-to-water-phase volume ratio of each DNSPE. Furthermore, the stability and composition of the emulsions were influenced by a multitude of factors.

### 5.2. Particle Size and ζ-Potentials

Particle size and ζ-potential are fundamental characteristics of DNSPEs. The methods currently published for determining the ζ-potential, which is directly determined by an instrument such as a Zetasizer Nano ZS90, exhibit a fair degree of consistency. Particle size is usually determined by one of two methods. One is to directly acquire the particle size parameters *d*_4,3_, *d*_3,2_, *d*_0.5_, and so forth, using a laser particle size analyzer based on dynamic light scattering technology. The other method is to determine the diameter of several hundreds of emulsion droplets from images captured by an optical microscope by utilizing specialized software and calculate the average particle size or *d*_4,3_ and *d*_3,2_ according to the following equation [33,38].
*d*_4,3_ = ∑n_i_*d*_i_^4^/∑n_i_*d*_i_^3^ *d*_3,2_ = ∑n_i_*d*_i_^3^/∑n_i_*d*_i_^2^(2)

The first method may include free nanocrystals that are not adsorbed on the surface of the emulsion droplets into the measurement range, so separation of free nanocrystals becomes necessary if a more precise particle size is desired. Nevertheless, the isolation of free nanocrystals from emulsion droplets without compromising their stability poses a significant challenge. Duffus [46] identified a potential error in the Mastersizer apparatus’s estimation of the droplet size of a rutin-stabilized sunflower oil emulsion. This error could have arisen from the device mistaking flocculated oil droplets for particle aggregates, which would be the result of the comparable refractive indices of rutin (1.77) and sunflower oil (1.47). The second method is greatly impacted by the visual representations of an emulsion as well as individual counting. There is no comparative study on whether the particle sizes determined by the two methods differ substantially. Additionally, further research is also needed to investigate whether sample dilution during particle size determination affects the measurement outcomes.

### 5.3. Stability

Stability is a fundamental criterion for evaluating the properties of DNSPEs. Presently, there are three prevalent methodologies for stability evaluation.
(1)Determine the changes in particle size of emulsion droplets subsequent to a specified duration of storage.(2)Determine the creaming index or coalescence index. After storing in vials for a certain period of time, unstable emulsions would develop an oil or water layer on the top or at the bottom of the vials, respectively.(3)The ratio of the measured height of the water phase or oil phase to the initial total height of the emulsion is considered the creaming index [46] or coalescence index [28], respectively. A greater creaming index or coalescence index indicates a diminished level of emulsion stability. Although this evaluation method is extensively employed, there are still some defects. It is a great challenge to precisely determine the volume of the oil layer, particularly when only a small amount of the oil phase is separated. It is also a great challenge to accurately determine the volume of the water layer in samples where the transition is gradual from the water layer to the emulsion layer without a clear demarcation.(4)Centrifugation stability. Centrifugation has been used to evaluate the stability of emulsions in many national pharmacopoeias. Due to its efficiency and simplicity of operation, this method is also frequently employed to assess the stability of DNSPEs. However, there are currently no unified and standardized centrifugation conditions. Based on the *Chinese Pharmacopoeia* (2020 edition) centrifugation at 1800× *g* for 15 min was selected for a DNSPE of puerarin [41], while centrifugation at 5000× *g* for 5 min and at 11,180× *g* for 10 min was utilized for DNSPEs of diosgenin [30] and luteolin [31], respectively. Furthermore, certain studies measured the quantity of the oil or water phase that precipitated from the DNSPE subsequent to centrifugation [30,31], while other studies examined the alteration in turbidity by measuring the absorbance of the emulsion sample at 500 nm [41]. A decrease in the absorbance value corresponded to a deterioration in the emulsion stability.

### 5.4. Interfacial Adsorption

For direct verification of the microstructure of DNSPEs, observation of the adsorption of drug nanocrystals at the oil–water interface of emulsion droplets is an indispensable characterization method. Scanning electron microscopy (SEM), inverted fluorescence microscopy (FM), and confocal laser scanning microscopy (CLSM) have been used frequently.

#### 5.4.1. SEM and Cryo-Scanning Electron Microscopy (Cryo-SEM)

SEM is used to observe not only the morphology but also the surface microstructure of emulsion droplets. The sample of the DNSPE should be dispersed onto the surface of a specific carrier medium, such as silica gel film, tin foil, or polycarbonate film. Following natural evaporation and gold spraying, the sample can be observed by an SEM. The presence or absence of drug nanocrystals adsorbing onto the surface of emulsion droplets can be determined through comparison with control emulsion droplets that do not contain nanocrystals. As shown in Figure 2, the surfaces of DNSPE droplets of silybin [36], puerarin [41], and quercetin [39] observed by SEM were uneven with numerous protruding edges, suggesting that drug nanocrystals might be adsorbed onto the droplets’ surfaces.

The sample chamber of a conventional scanning electron microscope is a high vacuum environment (10^−3^ Pa), which requires the observed sample to be desiccated and non-volatile. Therefore, the emulsion must be dehydrated prior to observation. However, the structure of the emulsion droplets may undergo alterations during the dehydrating process, which would prevent the accurate determination of their structure and particle size. In order to surmount this limitation, cryo-SEM can directly observe liquid and semi-liquid samples by employing ultra-low-temperature freezing sample preparation and transmission technology. Following freezing at an extremely low temperature (as low as −140 °C) and coating with gold/carbon spraying, the sample is transferred via the freezing transport system to the cold stage of the electron microscope, where the temperature can drop to −180 °C, for observation. Compared with drying via natural volatilization, freeze drying induces a comparatively minor alteration in the structure of emulsion droplets. This technique has been implemented to observe the structures of DNSPEs of curcumin [28] and rutin hydrate [46].

#### 5.4.2. FM

For insoluble drugs with their own fluorescence, such as puerarin and curcumin, FM can be used to directly observe the drug nanocrystals adsorbed on the surface of the emulsion droplets. As shown in Figure 3, a conspicuous circle of yellow-green autofluorescence of puerarin appeared clearly on the surface of emulsion droplets when a DNSPE of puerarin was observed by FM, confirming the adsorption of puerarin nanocrystals at the emulsion droplet interface [41].

#### 5.4.3. CLSM

Compared with FM, CLSM is more frequently used in the microstructure characterization of DNSPEs due to its significantly greater magnification. CLSM can be used for the direct observation of the autofluorescence of drug nanocrystals adsorbed on the surface of emulsion droplets, as illustrated in Figure 4.

For drugs that do not emit autofluoresce, some special methods can be used to render them fluorescent before observation. One is the chemical reaction method. Ginkgo biloba extract contains a large number of flavonoids, which can react with AlCl_3_ to produce fluorescent substances. Based on this principle, Yang et al. [35] mixed ginkgo biloba flavonoid-stabilized DNSPE with an equal volume of AlCl_3_ ethanol solution, and then observed the structure of the emulsion droplets at an excitation wavelength of 488 nm by CLSM. A red fluorescence of flavonoids stained by AlCl_3_ was observed around the emulsion droplets, suggesting some flavonoid particles were coated at the surface of the oil droplets, thus strongly confirming the formation of a Pickering emulsion.

Fluorescence labeling is another method. Nile red and Nile blue are the most popular fluorophores, used to label the oil phase and the aqueous phase, respectively. The fluorescence of Nile red in the oil phase and Nile blue on solid particles is observed at 488 nm and 633 nm, respectively. By combining images captured at two distinct wavelengths, it is possible to discern the adsorption of solid particles onto the surface of oil droplets. The specific procedures vary among studies. Some studies dissolved Nile red and Nile blue in the oil and aqueous phases, respectively, prior to preparing the DNSPE [30]. Others supplemented the prepared DNSPE with a 0.01% propylene glycol solution of Nile red and 0.1% aqueous solution of Nile blue for staining before observation [68,69]. For CLSM observation of Pickering emulsions stabilized by macromolecular particles, more fluorophores were used. For instance, fluorescein sodium (1 mg/mL) was used to label zein nanoparticles [70], fluorescein isothiocyanate (1 mg/mL, dissolved in propylene glycol) was used to stain surimi particles [71] and chitosan particles [72], calcofluor white was used to stain R-chitin [73], and acridine orange (1.0 mg/mL, in isopropyl alcohol) was used to stain chitin nanoparticles [23].

Strictly speaking, the substance detected by CLSM subsequent to the dispersion of the fluorophore in water is the fluorophore itself, rather than solid particles. It is necessary to ascertain that the fluorophore is capable of binding to solid particles and that any unbound fluorophore has been removed prior to CLSM observation. Only on this basis can the solid particles be faithfully represented by the fluorescence of the fluorophore. Regrettably, there are currently no studies providing supporting evidence of this.

#### 5.4.4. Drug Distribution and Existing Forms

Theoretically, upon mixing the drug nanocrystal suspension with the oil phase, part of the drug is dissolved in the oil phase in the form of molecules. Furthermore, the nanocrystals may also be freely dispersed in the water phase in addition to being adsorbed on the surface of the oil droplets. The existing forms and distribution of drugs significantly influence their fate and effectiveness once they enter the body. As a result, it is imperative to study the distribution of drugs across different phases of DNSPEs.

Luo et al. [45] centrifuged an n-hexane emulsion stabilized by flavonoid particles including rutin and naringin at 4000× *g* for 20 min. Following this, the aqueous phase was isolated and the concentration of free flavonoids in water was determined. Comparing the adsorption of drug nanocrystals in DNSPEs is possible with this method when the total drug concentration is constant.

Yang et al. [34] centrifuged a Pickering emulsion stabilized by ginkgo biloba leaf extracts at 12,000× *g* for 30 min. The transparent water layer at the bottom was collected carefully with a syringe, and the total quantity of flavonoids was measured. The surface coverage of ginkgo flavonoid compounds calculated according to the following formula was 40.01 ± 1.13 mg/m^2^. However, the extent to which the emulsion droplet surface was coated with nanocrystals, or the percentage of the emulsion droplet surface that was covered by solid particles, could not be determined using this technique.
*Γ =* (*C*_adsorbed_ × *d*_3,2_)/(*φ* × 6) = [(*C*_emulsion_ − *C*_sublayer_) × *d*_3,2_ ]/(*φ* × 6)(3)
where *Γ* (mg/m^2^) is the concentration of flavonoids on the surface, *φ* is the volume ratio of the oil phase in the emulsion, *C*_adsorbed_, *C*_emulsion_, and *C*_sublayer_ (mg/mL) are the concentrations of flavonoid components in each phase layer after centrifugation, and *d*_3,2_ (mm) is the average specific surface particle size of the emulsion droplets.

Alternative computational techniques can be employed to ascertain the adsorption of micromolecular solid particles onto Pickering emulsion droplets. For example, the interfacial coverage was used to assess the adsorption of cellulose nanotubes within a Pickering emulsion. The calculation equation was as follows [23].
$$C=100\times \frac{m_{p}D_{3,2}}{6h\rho V_{\mathrm{oil}}}$$
where *m*_p_ is the mass of cellulose adsorbed (kg), *D*_3,2_ is the surface area diameter (m), *h* is the thickness of the cellulose adsorbed layer (m), *ρ* is the density of cellulose nanocrystals, and *V*_oil_ is the volume of the oil phase in the emulsion.

Li [74] evaluated the adsorption of bacterial cellulose nanofilaments on the surface of dodecane emulsion droplets in another way. The surface coverage (C) of the oil droplets was calculated using the following equation:$$C=\frac{M_{P}R}{3h\rho V_{\mathrm{oil}}}\times 100\%$$
where *M*_p_ is the mass of cellulose in the emulsion (g), *h* is the thickness of the cellulose adsorbed layer which can be observed by AFM, *ρ* is the density of the cellulose nanocrystals (1.6 g/cm^3^ in this study), and *V*_oil_ is the volume of the oil phase in the emulsion.

In order to precisely quantify the quantity of protein adsorbed on the surface of the oil droplets, Chevallier et al. [75] used sodium dodecyl sulfate-polyacrylamide gel electrophoresis (SDS-PAGE) to recover proteins adsorbed on the surface of the oil droplets. These methods provide a reference for quantitative analysis of drug nanocrystal adsorption in DNSPEs. Whether they are suitable and how to apply them to DNSPEs is still subject to investigation.

Wang et al. [60] separated different phases of DNSPEs by centrifuging a fresh DNSPE of puerarin and tanshinone IIA at 1800× *g* for 15 min to remove free nanocrystals first, followed by centrifugation at 4 °C, 50,000 rpm for 1 h. The volume and drug concentration of the oil phase in the upper layer and the clear aqueous phase in the lower layer were precisely determined after they were separated. The adsorbed drug amount in the intermediate layer at the oil–water interface could be calculated by subtracting the drug amounts present in both the water and oil phase from the total amount in the stable DNSPE. The rate of drug distribution in each phase could ultimately be calculated. The results showed that 70.3% of the puerarin was dissolved in the water phase and 29.3% was adsorbed on the surface of emulsion droplets in the form of nanocrystals. Due to the nearly insoluble nature of puerarin in the oil phase of Ligusticum chuanxiong oil, its amount in the oil phase was less than 1%. Conversely, tanshinone IIA exhibited high solubility in the oil phase but negligible solubility in water, so approximately 26% of tanshinone IIA was dissolved in the oil phase and approximately 74% was adsorbed on the interface of the emulsion droplets. By employing this approach, one can acquire insights into the drug distribution within a DNSPE, thereby facilitating comprehension of its microstructure.

Overall, the existing research methods for measuring drug distribution in DNSPEs all rely on the critical step of separating the water phase from the oil phase via high-speed centrifugation. The accuracy of the methods is still questionable, especially for drugs with high solubility in the oil phase, due to the typically small quantity of the oil phase in DNSPEs and the challenge of precisely determining their volume. Furthermore, the centrifugation speed and time need to be optimized for a particular DNSPE. Distinguishing the nanocrystals free in the aqueous phase from the nanocrystals adsorbed on the surface of emulsion droplets presents an additional difficulty associated with this method.

## 6. In Vivo Fate of DNSPEs

As a drug delivery system, the fate of DNSPEs after entering the human body is closely related to their effectiveness. Present studies on the in vivo fate of Pickering emulsions focuses primarily on the food field. The impact of macromolecule solid stabilizers such as protein, starch, and cellulose on lipid digestion and the bioaccessibility of drugs loaded in the oil phase was investigated using a simulated gastrointestinal tract model [76]. The prevailing procedure involved incubating emulsions in succession with simulated saliva fluid, simulated gastric fluid, and simulated intestinal fluids. The alterations in the emulsion particle size and ζ-potential, the structure of the emulsion droplets, the release kinetics of the free fatty acids, and the bioavailability of the loaded drugs were assessed [16,22,68].

The gastrointestinal fate of some DNSPEs has been investigated utilizing this method. Wang et al. [31] investigated the stability of pine nut oil Pickering emulsions stabilized by luteolin micro-nano particles in simulated gastrointestinal fluid. The emulsion was added dropwise into simulated gastric fluid and simulated intestinal fluid, and stirred at 37 °C for 2 h and 6 h, respectively. The resulting variations in particle size and *ζ*-potential were measured. The results showed that a small amount of luteolin might detach from the surface of the emulsion droplets in the simulated gastric and intestinal fluids, resulting in a slight increase in particle size and ζ-potential. The relative turbidity of the emulsion was above 90%, suggesting that the DNPSE maintained a commendable stability when exposed to gastrointestinal fluid.

A puerarin DNSPE, however, showed a different stability in gastric fluid from the luteolin DNSPE. After the puerarin DNSPE was incubated with simulated gastric fluid for a specified period of time, the samples became transparent. This phenomenon became more pronounced with increasing amounts of simulated gastric fluid and incubation time. Meanwhile, the particle size of the emulsion droplets decreased, and the green fluorescence circle of puerarin nanocrystals surrounding the surface of the emulsion droplets became weaker. These results suggested that the simulated gastric fluid might have caused some damage to the microstructure of the DNSPE [77]. Subsequently, the samples underwent further incubation in simulated intestine fluid through pH adjustment to 7.5 and addition of bile extract and pancreatic enzyme. After 2 h, alongside the presence of spherical emulsion droplets, a considerable quantity of drug crystals with a particle size of approximately 0.5–1 μm was also observed [77].

The absorption of DNSPEs in various intestinal segments could be investigated using an in situ perfusing experiment in rats. Wang et al. [78] studied the intestinal absorption characteristics of a puerarin DNSPE using a single-pass intestinal perfusion model. The absorption rate constant (*K*_a_) and the intestinal apparent permeability coefficient (*P*_app_) decreased in the following order: duodenum, jejunum, ileum, and colon segments. Notably, the duodenum exhibited a higher value compared to the jejunum and ileum (*p* < 0.05), and this was even more significant compared to the colon (*p* < 0.01).

The absorption and transport mechanism of DNSPEs across intestinal epithelial cell membranes could be studied using the Caco-2 cells model. Wang et al. investigated the absorption mechanism of a puerarin DNSPE using Caco-2 cell uptake and transport experiments. It was determined that uptake of the DNSPE involved both passive and active transport mechanisms, in addition to endocytosis mediated by lipid rafts and caveolin [79]. This mechanism was also confirmed by a study on DNSPE loaded with the main components of the Tongmai formula. The uptake of pharmacodynamic components such as puerarin, ferulic acid, ligustilide, and tanshinone IIA, was significantly enhanced compared with the physical mixture of the nanocrystal suspension and oil [42]. This suggested that the adsorption of drug nanocrystals onto the surface of micron-sized oil droplets, resulting in the formation of a “micro” and “nano” synergistic microstructure, could potentially play a crucial role in facilitating the cellular uptake and transport of drugs for DNSPEs.

## 7. Prospects

As unique Pickering emulsions, DNSPEs have promising potential applications in drug delivery. They have been widely used to delivery poorly water-soluble drugs via oral, topical, and even injectable administration. However, there remain numerous critical concerns pertaining to their theory and implementation that require thorough investigation.

First, some fundamental concerns concerning the preparation and stability of DNSPEs should be explored extensively. For example, what methods can be employed to precisely quantify the actual contact angle of drug nanocrystals, rather than raw materials, at the oil–water interface, in order to reveal the genuine correlation between the three-phase contact angle of nanocrystals and the emulsion stability? How might drug nanocrystals influence the tension at the interface between oil and water? What is the relationship between nanocrystal size and emulsion droplet size? Is this relationship affected by the characteristics of the drugs or oils? The impact of various nanocrystal shapes (e.g., spherical, elongated columns, flakes, etc.) on the efficacy of DNSPEs remains unknown. The drug loading capacity of DNSPEs is limited by the saturated interfacial adsorption. Is it feasible to develop a supersaturated DNPSE containing free nanocrystals to increase the drug loading capacity? In terms of structural characterization, new methods are required to accurately distinguish free nanocrystals dispersed in the continuous phase from those adsorbed on the surface of the emulsion droplets, to quantify precisely the distribution of the drug in each phase of a DNSPE, and to determine the extent to which the surfaces of emulsion droplets are covered by nanocrystals.

The research on DNSPEs is conducted in the laboratory presently, and substantial progress remains to be made before industrial production and market sales. The current quality standards of DNSPEs are not perfect, and the methods and conditions employed for stability assessments diverge, thereby failing to satisfy the prerequisites of novel drug development. Assuring that DNSPEs produced at the industrial scale and in laboratories have identical particle sizes and interface adsorption rates is a formidable challenge. Are DNSPEs administered as an emulsion, are they amenable to solidification and formulation into capsules or tablets? Although spray drying and freeze drying have been frequently used to solidify Pickering emulsions [80,81], there are only a few studies on the drying of DNSPEs. Up to now, only spray-drying a puerarin DNSPE with HP-β-cyclodextrin as a carrier [82] and freeze-drying a curcumin DNSPE with mannitol as a lyoprotectant [37] have been reported. Considering that the stabilizers adsorbed at the interface of DNSPE droplets are different from macromolecules, further research is warranted to ascertain whether the carriers used for spray drying the lyoprotectants used for freeze drying for traditional Pickering emulsions are still suitable for DNSPEs. The impact of the drying process on the adsorption of drug nanocrystals in DNSPEs, as well as the subsequent alterations in behavior and efficacy of drugs in vivo have not yet been studied. These are crucial considerations that must be taken into account during the development of novel pharmaceuticals based on DNSPEs.

The alterations of DNSPEs in various physiological environments under different administration routes are crucial for their clinical application. Are nanocrystals capable of desorbing from the surface of the emulsion droplets in physiological environments? How long can the emulsion droplets maintain their structural integrity? Will there be substantial alterations in the characteristics of nanocrystals, including morphology, size, and charge? Determining the absorption mechanisms of drugs dispersed in various phases of DNSPEs is of utmost significance in order to ascertain which phase distribution is more conducive to promoting therapeutic effects and minimizing toxic side effects. These issues dictate the suitability of DNSPEs as a secure and effective drug delivery system.

The difficulty in understanding the above issues lies in how to accurately monitor the evolution of drug nanocrystals, particularly in differentiating them from dissolved drug molecules. In recent years, an environment-responsive fluorescent probe hybrid technology has been successfully used to study the ex vivo and in vivo fate of drug nanocrystals [83]. This technology made distinguishing nanocrystals from free drugs dissolved in mediums possible due to the probe’s fluorescence-quenching phenomenon in various environments. P2 and P4 are fluorescent probes with an aggregation-induced quenching (ACQ) effect. They emit fluorescence when hybridized to nanocrystals as molecules. Once the nanocrystals are dissolved, the probes are released into water and rapidly aggregate as a consequence of their inadequate solubility in water, resulting in complete quenching of the fluorescence. The P2 probe has been hybridized with quercetin [84] and curcumin [85] nanocrystals. The in vivo distribution and elimination of drug nanocrystals subsequent to intravenous injection were observed by an in vivo imaging technology. Tetraphenylethene (TPE) and tetrakis (4-hydroxyphenyl) ethylene (THPE) are other fluorescent probes with the aggregation-induced emission (AIE) effect. The emission of fluorescence occurs when they are hybridized in nanocrystals and their molecular motion is restricted. The dissolution of nanocrystals results in the release and subsequent dissolution of the probes in water, which enables unrestricted motion of the probe molecules and extinguishes the fluorescence. By hybridizing TPE with paclitaxel nanocrystals, Li’s group demonstrated that intact nanocrystals of paclitaxel could be internalized by HT-29 and KB cells [86]. The molecular dynamics of the uptake and exocytosis of THPE nanocrystals by KB cells in their own crystalline and molecular forms was further investigated [87]. These investigations could potentially serve as the groundwork for further examining the in vivo fate and in vitro structure of DNSPEs.

As a new type of Pickering emulsion, DNSPEs have attracted more and more attention in recent years due to their ability to reduce potential safety risks posed by surfactants or certain solid particles and their high drug loading capacity. It is anticipated that DNSPEs, as an innovative drug delivery system for insoluble drugs, will continue to advance and eventually be implemented clinically in the future, as more knowledge is acquired regarding the underlying theories and practical challenges.

## Figures and Tables

**Figure 1 pharmaceutics-16-00293-f001:**
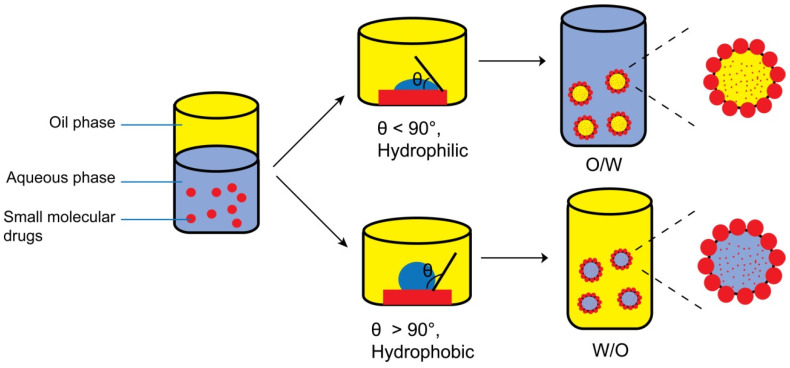
The relationship between wettability (three-phase contact angle, *θ*) of the solid particles and type of Pickering emulsion.

**Figure 2 pharmaceutics-16-00293-f002:**
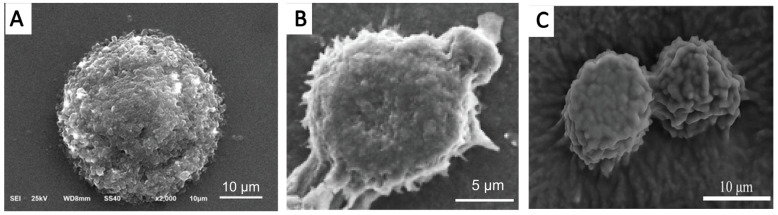
Scanning electron micrographs of DNSPE droplets of silybin (**A**), puerarin (**B**), and quercetin (**C**). Adapted from Refs. [36,39,41] with permission, Copyright © 2016 Elsevier B.V., Copyright © 2018 MDPI, and Copyright © 2022 MDPI, respectively.

**Figure 3 pharmaceutics-16-00293-f003:**
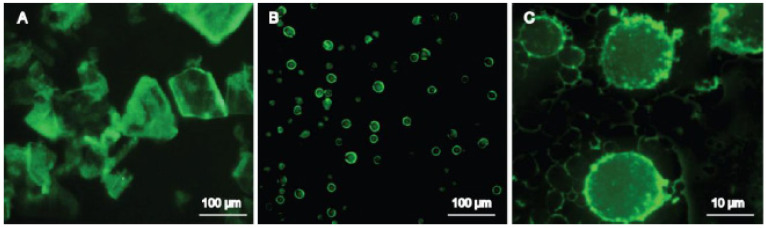
Fluorescence micrographics of (**A**) crude material and (**B**,**C**) DNSPE of puerarin. Adapted from Ref. [41] with permission. Copyright © 2018 MDPI.

**Figure 4 pharmaceutics-16-00293-f004:**
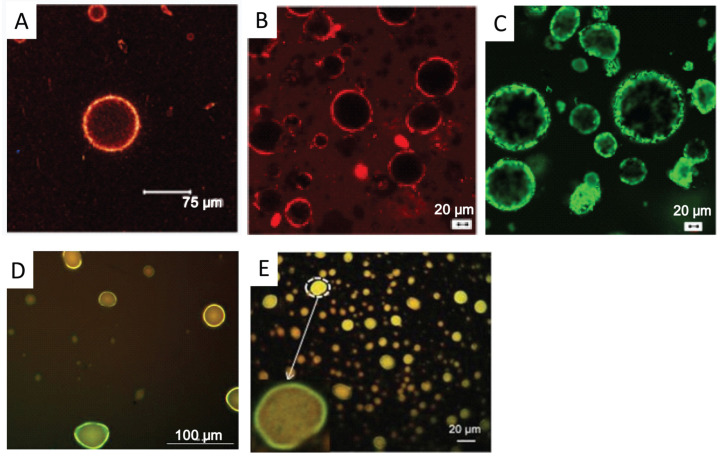
CLSM images of emulsions stabilized by tiliroside (**A**), curcumin (**B**), quercetin (**C**), luteolin (**D**), and puerarin (**E**) particles. The brightness in the images is caused by autofluorescence of drugs. (**A**) was adapted from Ref. [68] with permission. Copyright © 2011 American Chemical Society. (**B**,**C**) were adapted from Ref. [38] with permission. Copyright © 2018 American Chemical Society. (**D**) was adapted from Ref. [31] with permission. Copyright © 2020 Society of Chemical Industry. (**E**) was adapted from Ref. [41] with permission. Copyright © 2018 MDPI.

**Table 2 pharmaceutics-16-00293-t002:** The operation and results of contact angle measurement in reported DNSPE studies.

Active Ingredient	Oil	Determination Operation	Results	Reference
Curcumin	High-oleic sunflower oil	Nanosuspensions were lyophilized and pressed into a tablet. Then, 5 μL of water was placed on the flat surface.	*θ*w/a: 59°	[28]
Curcumin	Soybean oil	A volume of 3 μL of water or oil was added to the compressed disc surfaces of drug crystals.	*θ*w/a: 73.4 ± 1.2° *θ*o/a: 11.9 ± 2.0°	[38]
Quercetin	Soy oil	Thick slices of lyophilized quercetin nanocrystals were placed into soybean oil, and 5 μL of water droplets were released onto the surface.	*θ*w/o: 65°	[40]
Ursolic acid	Rapeseed oil	Glass substrates coated with ursolic acid films were placed on top of an oil subphase. Then, a water droplet (~5 μL) was released from the tip of a high-precision syringe onto the surface of the UA film, allowing it to be absorbed on the UA film.	*θ*w/o: 156.4 ± 0.1°	[33]
Puerarin	Ligusticum chuanxiong oil and Labrafil M 1944 CS (9:1, *v*/*v*)	A tablet of lyophilized puerarin crystal was immersed in oil for 30 min. Then, 2 μL of water was dropped on the interface.	*θ*w/o: 81 ± 4°	[41]
Dihydromyricetin	Medium-chain triglyceride	DMY was compressed into a film and placed into MCT. A water drop was added on the film surface.	*θ*w/o: 118.8°	[32]
Diosgenin	Canola oil	Diosgenin was compressed into films. Then a water drop around 3 μL was placed on the film surface using a micro syringe.	*θ*w/a: 139.6°	[30]
Rutin hydrate	Sunflower oil	A volume of 7.5 μL of water or oil droplets was spotted onto compressed particle dick/pellet surfaces.	*θ*o/a: 77° *θ*w/a: 34°	[46]
Naringin			*θ*o/a: 34° *θ*w/a: 25°	

Note: *θ*w/a means contact angle in air–water, *θ*o/a means contact angle in air–oil, *θ*w/o means contact angle in water–oil.

## Data Availability

Not applicable.

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
