# Peer review of "Progress of Drug Nanocrystal Self-Stabilized Pickering Emulsions: Construction, Characteristics In Vitro, and Fate In Vivo"

_pharmaceutics, 2024, doi:10.3390/pharmaceutics16020293_

Round 1
Reviewer 1 Report
Comments and Suggestions for Authors
The current manuscript by Zhang et al is a review on the use of drug particles as stabilizers of Pickering emulsions, considering all aspects of Pickering stabilization: particle size, charge, wettability, concentration, and methods for preparation. It also summarizes some of the main techniques for the characterization of the emulsions prepared. The review is short but concise, as it summarizes 90 articles considering the topic. I recommend its publication after the authors address my questions and comments below:
The review does not offer much to question, besides the lack of clearly presented mechanisms for increased drug loading and kinetics of dissolution for these drug-Pickering droplets. Is there clear evidence to prove Pickering emulsion is better than an emulsion with dissolved crystals in the droplets? How much better is it? What are the mechanisms?
It would be nice for the authors to present some hypotheses if clear mechanisms are missing. It seems that the improvement in the loading in the emulsions might be related to (1) some crystals dissolving in the oil phase of the emulsions, (2) increasing the surface area of the drug crystals through homogenization, and (3) enhanced exposure of particles due to interlocking at the interface that affects their kinetics of dissolution.
Author Response
Dear reviewer,
Thank you for providing us the valuable comments and suggestions on our manuscript. We value your comments and suggestions and address each of the points in sequence below. All the modified words or sentences have been incorporated in the revised manuscript and these modifications have been marked in red.
Best regards
Jifen Zhang
The review does not offer much to question, besides the lack of clearly presented mechanisms for increased drug loading and kinetics of dissolution for these drug-Pickering droplets. Is there clear evidence to prove Pickering emulsion is better than an emulsion with dissolved crystals in the droplets? How much better is it? What are the mechanisms?
It would be nice for the authors to present some hypotheses if clear mechanisms are missing. It seems that the improvement in the loading in the emulsions might be related to (1) some crystals dissolving in the oil phase of the emulsions, (2) increasing the surface area of the drug crystals through homogenization, and (3) enhanced exposure of particles due to interlocking at the interface that affects their kinetics of dissolution.
Reply: We are very grateful for your valuable suggestions. DNSPE is a novel Pickering emulsion and many issues remain to be investigated. Considering the lack of sufficient research to reveal the mechanisms clearly, we have added some hypotheses to the revised manuscript based on your suggestions as follows.
“Compared with traditional emulsions stabilized with surfactants, drugs in DNSPE were not only dissolved in the oil phase but also adsorbed at the oil-water interface of the droplets. Differences in the structure lead to different drug loading and dissolution kinetics. It seems that the improvement in drug loading and dissolution might be related to (1) some crystals dissolving in the oil phase of the emulsions, (2) increasing the surface area of the drug crystals through homogenization, and (3) enhanced exposure of particles due to interlocking at the interface that affects their kinetics of dissolution. The clear mechanism remains to be further explored.”

Reviewer 2 Report
Comments and Suggestions for Authors
The Authors presented an overview of the drug self-stabilized Pickering-type emulsions. The topic is suitable for the Journal's aims and scopes. Furthermore, the references are quite new. My recommendation is to accept after minor improvements.
Detailed comments are listed below:
- please check the section numbering, and highlight that the first section is an introduction
- materials and methods - please add this section where you present the data selection methodology i.e.: key words selection, time period, content, etc.
Consider shifting the tables after the materials and methods part, which you will include.
Section "Particle size of nanocrystals - please specify why you presented here structures whose sizes are bigger than 100nm (check the nanostructure definition). A more suitable description will be submicron-sized structures.
Please add more figures that will show real examples of the prepared emulsion systems - I'm sure that there are a lot of images that you can use i.e. in Molecules, IJMS, Biomolecules, etc.
Author Response
Reply:Thanks for your suggestion. We highly valued your suggestions. Please allow us to explain why no more figures were added. This review focused on Pickering emulsion stabilized by small molecular active particles, whose appearance and optical morphology of emulsion droplets were not significantly different from other emulsions. The valuable figures should be those showing adsorption of small molecular active particles on the oil-water interface of emulsion droplets. However, there were not much researches in this area. We have selected representative and clear figures to be included in the manuscript. There were also some other images, but they were not clear and could not clearly indicate the adsorption of particles at the droplet interface. In addition to the expensive copyright fees, they were not added in the revised manuscript. We had referred them in our review in case that interested readers could find them in the original papers. We sincerely anticipate your understanding.
Some figures which were not included were presented iin the following file.

Reviewer 3 Report
Comments and Suggestions for Authors
I think this is an easy-to-understand summary of the nano-particulation method of pharmaceuticals, focusing on the pickering emulsion method.
Author Response
Dear reviewer,
Thank you for your positive comments.
Best regards
Jifen Zhang
Reviewer 4 Report
Comments and Suggestions for Authors
Although this paper is interesting, it should be improves according to following lines:
1- Additional column should be added in Table 1 regarding pharmaceutical properties of active ingredients in column 1. What drug delivery properties were achieved by researchers?
2- Preparation methods of DNSPE should be summarized in a table.
3- At Section 3.1, Different contact angles of DNSPE should be summarized in a table.
Author Response
Dear reviewer,
Thank you for providing us the valuable comments and suggestions on our manuscript. We value your comments and suggestions and address each of the points in sequence below. All the modified words or sentences have been incorporated in the revised manuscript and these modifications have been marked in red.
Best regards
Jifen Zhang
- Additional column should be added in Table 1 regarding pharmaceutical properties of active ingredients in column 1. What drug delivery properties were achieved by researchers?
Reply:Thanks for your suggestion. We have added pharmaceutical properties of active ingredients and drug delivery properties in Table 1.
- Preparation methods of DNSPE should be summarized in a table.
Reply:Preparation methods of reported DNSPE have be summarized in Table 1.
- At Section 3.1, Different contact angles of DNSPE should be summarized in a table.
Reply:The contact angle measurement methods and results in existing DNSPE studies have been summarized in Table 2.
